# Use of Virtual Reality Technology to Support the Home Modification Process: A Scoping Review

**DOI:** 10.3390/ijerph182111096

**Published:** 2021-10-21

**Authors:** Na-Kyoung Hwang, Sun-Hwa Shim

**Affiliations:** 1Department of Occupational Therapy, Seoul North Municipal Hospital, Seoul 02062, Korea; occupation81@gmail.com; 2Department of Occupational Therapy, College of Medical Science, Jeonju University, Jeonju 55101, Korea

**Keywords:** collaboration, education, home environment measurement, occupational therapist

## Abstract

Healthcare is a field in which the benefits of virtual reality (VR), such as risk-taking without consequences, direct experience, and service outcome prediction, can be utilized. VR technology has been used to help clients face environmental barriers by implementing a home environment in virtual reality without a home visit by an expert. This scoping review was conducted to identify the areas and implementation methods of the home modification process supported by VR technology. Twelve studies met the research criteria. The following three types of tools supported by VR technology for the home modification process were identified: educational tools for clients or specialists, home environment measurement tools, and intermediaries for decision making and collaboration between clients and specialists. Most of the studies reported positive results regarding the usability and acceptability of the technology, but barriers have also been reported, such as technical problems, inappropriate population groups for technical use, cost-related issues, the need for training, and fear that the technology could replace home visits. Thus, VR technology has potential value in the home modification process. However, for future clinical applications, additional studies to maximize the benefits of these VR technologies and address the identified problems are required.

## 1. Introduction

Home modification, as part of occupational therapy (OT) interventions, is a general compensatory strategy used to improve occupational performance and reduce environmental barriers in the home environment of clients with functional limitations [1,2]. It has traditionally been implemented based on the “Person–Environment–Occupation” model, which is a conceptual framework for OT [3]. Occupational therapists provide a home environment suitable for individuals by supporting a virtuous cycle in which people, activities, and spaces are mutually supported and utilized [4]. OT-led environmental assessment and modification generally comprise comprehensive assessments of the individual, their activities of daily living, and their environment and strategic interventions focused on these areas. Strategic interventions include the prescription of assistive technology and adaptive training, material adaptation such as applying non-slip strips on stairs, behavioral adaptation such as refraining from hazardous activities, and structural modifications such as installing stair lifts [5]. Home modification services reduce the caregiver burden of client care [6], delay the admission of the client to a facility [7], and contribute to the reduction in falls [8]. Studies have also reported that home modification by occupational therapists increases clients’ self-awareness of activity performance [9] and their acceptance of the results of home modification [10].

Home modification services are organized in various ways in different countries; however, they share the involvement of multiple stakeholders such as paying authorities, builders, and interdisciplinary healthcare team members [2]. To modify the home environment optimally, it is necessary understand the environmental barriers in the home and the structural changes required to solve them, the use of assistive technology, and changes in the client’s occupational performance, and to collaborate with stakeholders to address these areas [11]. The lack of communication between stakeholders involved in the home modification process can impede the smooth progress of the process and result in environmental modifications that do not reflect the client’s needs and desires, leading to the client’s maladjustment to the environment and a decrease in their satisfaction [12,13]. In addition, 2D drawings of spatial design, photos, or online image prints of assistive equipment, which have been primarily used for coordination and collaboration with stakeholders during the home modification process, are limited in their ability to help the stakeholders fully understand the environmental modification. In particular, it may be difficult for clients to imagine or surmise the application of assistive equipment when only image materials are provided, without a demonstrative experience [14].

Virtual reality (VR) is an immersive and interactive technology that marks new milestones in the ways people interact with their environment and envision new approaches in their relationship to reality [15]. The immersive environments created by VR technology, such as 3D virtual worlds or games, are interesting because of their potential to enhance learning through situational experiences, various perspectives, or knowledge transfer [16]. VR has been gradually expanded and applied to various fields, such as defense, medical care, architecture, and education, and related content is being actively developed [17]. In healthcare, attempts to actively introduce VR to rehabilitation treatment, surgery/treatment/medical training, and emotional management have increased remarkably [17]. The value of VR lies in enabling users to directly experience the targeted situation without experiencing the risks or errors that may occur in real situations [18]. In particular, the healthcare field is an area where the advantages of VR, such as opportunities for risk-taking and direct experience, are usefully applied [19]. Simulation is considered a standard method in healthcare education, and technology-enhanced simulation (TES) can be implemented using VR [20]. TES enables majors in healthcare-related fields to acquire knowledge and skills without harming clients or being in high-risk situations and to apply and practice the theory learned [21]. Moreover, instructors can provide feedback on practice in a safe and controlled setting [17].

E-health technologies, such as VR, enable user-centric access to healthcare. Clients receiving healthcare services can practice and experience target behaviors, and VR is useful in predicting service outcomes [19]. Recently, VR technologies have been used to help clients face environmental barriers by realizing the home environment without home visits. VR-based home modification software facilitates the cooperation between the client and expert in the home modification process by enabling the client to interact with various virtual home environments. It further increases the client’s insight into modification and helps home adaptation [22]. Moreover, these technologies offer the potential to increase access to home modification services for people in rural areas, reduce home assessment costs, and improve the quality of the process by supporting home assessment [1].

Ninnis et al. [23] reviewed existing technologies, such as digital photos and teleconferencing equipment, and new applications to support home assessments, but areas of home modification process supported by VR technology and the ways the technology is used have not yet been synthesized. The purpose of this study is to examine the areas and implementation methods of home modification processes supported by VR technology and explore the outcomes and barriers to VR technology application in home modification.

## 2. Materials and Methods

This review of VR technology use for home modification has applied a scoping review approach. According to the Arksey and O’Malley framework [24], scoping review has the following logical flow stages: (1) identifying research questions; (2) identifying relevant studies; (3) study selection; (4) charting the data; (5) collating, summarizing, and reporting the results; and (6) an optional consultation stage. The final step of the framework was not performed in this review because of the exploratory nature of the review and lack of stakeholder involvement.

### 2.1. Stage 1: Identifying Research Questions

The following three research questions present a broad scope of VR technology use for the home modification process:Which areas of the home modification process are supported by VR technology and how is the technology implemented?How are the VR technologies evaluated in terms of usability, acceptability, and participant outcomes?What barriers to using the identified VR technologies were identified?

### 2.2. Stages 2 and 3: Identifying Relevant Studies and Study Selection

Related studies were identified according to the Preferred Reporting Items for Systematic Reviews and Meta-Analyses (PRISMA) guidelines [25] (Figure 1). We searched four databases (MedlineComplete, Embase, CINAHL, and PsycINFO) to identify relevant studies on the use of VR technology for the home modification process published between 2000 and 2021. Moreover, the reference lists of the studies were searched to identify additional relevant studies. We also considered grey literature, which may be published in non-scientific journals, such as technology developments and case studies. Our basic search included search terms related to “virtual reality”, “home modification”, and “occupational therapy”. The following keywords were used: “virtual reality”, “mixed reality”, “augmented reality”, “3D” and “home modification”, “residential modification”, “environmental modification”, “home adaptation”, “home visit”, “pre-discharge home visit” and “occupational therapy”, “assessment”, “evaluation”, “assistive technology”, “education”, and “collaboration”.

The inclusion criteria for the research articles included research emphasis on the use of VR technology for home modification and availability of quantitative or qualitative data in full-text English. We excluded protocol studies, dissertations, theses, conference abstracts, and viewpoints. We did not limit the subjects of the study (e.g., to the elderly and patients) or the specific purposes of the use of VR technology (e.g., compensatory approach for home adaptation of clients with functional impairment), and we included studies in which VR technology was used in the home modification process. This review also included interventions using VR technology that are considered perspectives of OT practice in the home modification process, even if the technology developer or provider was not an occupational therapist.

The study titles and abstracts were examined after the initial search, and full texts of eligible studies were obtained. The manuscripts were searched for the presence of inclusion and exclusion criteria. A consensus to include each study in the scoping review was reached between the authors.

### 2.3. Stages 4 and 5: Charting the Data and Collating, Summarizing, and Reporting the Results

Data on the author, publication date, study design, home modification area, purpose, method, participants, main outcomes, and barriers were extracted from each article. One of the two authors performed data extraction, and the other verified that the data synthesis strategy was followed. A qualitative synthesis of the studies selected for this review was conducted. A tabular form was organized to compare the thematic information extracted from each study, and after analyzing the themes and barriers, textual descriptions were created for each study. The identification and extraction of thematic information was undertaken based upon areas of the home modification process supported by VR technology and the barriers in practice. Textual descriptions of each study are followed by the tabulation of results, grouping studies according to areas of VR technology use for home modification, the purpose and results of each study, how the technology was implemented, and VR technology barriers in practice. In Section 3, the themes and summaries of the studies are organized by their research questions, and the findings related to each question are discussed.

## 3. Results

The research literature on the use of VR technology for home modification was sparse at the time of this review, with only 12 articles meeting the inclusion criteria (Table 1). Figure 1 shows a PRISMA chart illustrating the search results and review selection.

### 3.1. Characteristics of the Included Studies

A total of 297 participants were included in the 12 studies. Sample sizes varied across the studies, ranging from 3 [26] to 69 participants [19]. In most studies, except for one [26], occupational therapists were involved as technology developers or providers. Most of the included studies investigated participant perspectives, such as usability, acceptability, and use of VR technology, and the data collection tools used were specifically designed for the studies but were not verified. A summary of the characteristics of the included studies is presented in Table 1.

### 3.2. Areas of the Home Modification Process Supported by VR Technology

Three VR-supported areas of the home modification process were identified from these studies. Four studies [19,27,28,29] used VR technology as an educational tool for clients or students of home modification, and three studies [22,30,31] used it as a tool to evaluate the home environment. The remaining five studies [26,32,33,34,35] used it as a tool to assist with optimal decision making and collaboration with clients in the home modification process.

### 3.3. VR as an Educational Tool for Clients (Stroke Survivors, People with Physical Disabilities, and the Elderly) and OT, PT Students

Threapleton et al. [27] applied a VR program designed as an educational tool that identified potential safety issues in the general home environment of stroke patients before they are discharged and informed of the need for assistive equipment and adaptations in the home. The study explored the acceptability, potential utility, and limitations of the VR program to occupational therapists and stroke survivors. The participants reported software-related technical issues, but they perceived the virtual home program as an effective visual means for the preparation and discussion of discharge planning.

Palmon et al. [28] evaluated the usability of the Habi-Test, which was designed to facilitate the planning, design, and assessment of optimal home and work settings for people with physical disabilities. Habi-Test addresses the need for environmental modifications by identifying barriers that limit the performance of tasks in the home environment, while the user navigates in a virtual home environment. Eight occupational therapists did not complete the task on the first attempt using the software, but after a few alternations, five out of the eight completed the task. The occupational therapists recommended continuous improvement and testing of the Habi-Test to address the frustration of the narrow field of view, difficulty navigating in the environment because of the narrow space, and difficulty in recognizing the current location.

Falls Sensei is a first-person 3D exploration game aimed at educating the elderly on external fall risk factors within the home environment. Users play a game to find all the risk factors present in each of the four areas (kitchen, bathroom, bedroom, and lounge and stairs) in the home. If they find a risk factor successfully, they are congratulated and a confirmation with an explanation of the risk factor is displayed on the screen. Money et al. [29] conducted usability tests for the program and investigated older adults’ perceptions and attitudes toward using the game in practice through a post-task interview. The participants reported “Good” levels of usability (systems usability scale score: 77.5/100) and positive attitudes in most items (*p* ≤ 0.05 for 9/10 items). In addition, the participants improved their awareness of home hazard detection and became more aware of future modifications required in their own homes.

Second Life^®^ is a web-based virtual environment used as an educational tool for clinical evaluation and intervention in the home environment. It provides undergraduate students with educational opportunities for home modification interventions outside of the classic classroom environment. Sabus et al. [19] applied the VR program to OT and physical therapy (PT) undergraduate students and investigated the utility of the program and the achievement of decision-making learning goals for home modification. As a result, the participants reported that the VR program supported learning about home modification and promoted collaboration between majors: PT students focused on functional movements during a client’s daily activities, OT students focused on compensatory approaches, such as installing contrasting stairs for clients with reduced vision, and environmental modification recommendations were derived through the collaboration between majors. In addition, the participants showed high levels of achievement in learning goals for home modification decision making, such as a client-centered approach, consideration of context factors, and appreciation of unintended consequences.

### 3.4. VR as a Tool for Home Environment Measurement

In other studies, VR programs were used to measure and evaluate the home environment, such as room size, furniture, and door width, for structural change to improve the accessibility of the client’s home and the selection of suitable assistive equipment. The potential for home measurement using an adapted commercial MagicPlan mobile application (MPMA) and laser distance measurer (LDM) was explored by Tsai et al. [30]. They investigated the feasibility and usability of the MPMA and LDM with lay participants and clinicians. MPMA allows users to create floor plans that can contain virtually inserted pieces of durable medical equipment (DME). MPMA with an LDM allows users to virtually determine whether the ordered DME will fit in the measured environment. The results showed that 77% of lay participants spent less than 60 min completing floor plans, and clinician participants completed virtual home evaluations within 5 min in 73% of the cases. In addition, both participant groups felt that using MPMA for home evaluations was useful and recommended it. However, the ease of use of the MPMA was scored as neutral by both groups.

The 3D measurement aid prototype (3D-MAP) proposed by Hamm et al. [31] was designed to support the elderly in directly measuring and recording installations in their homes as part of the process of providing assistive equipment. For accurate measurement, an audio guide and 3D visual model, which guided the measurement of five installations (bed, bathtub, toilet, chair, and stairs) related to external factors of falls in the home environment, were provided. Although the need for some changes to the use of 3D-MAP was identified, most participants perceived this app as a useful tool to help with the measurement and reduce the time for the auxiliary equipment provision process. Moreover, the app showed good levels of usability and strong agreement among the participants, especially in terms of usability and learnability.

MapIt, developed by Guay et al. [22], allows a room to be scanned quickly and simply, produces a 3D representation of a person’s home, and explores home adaptations to enhance a person’s occupational engagement. Guay et al. documented the development of the app prototype and investigated the stakeholder views of the app’s acceptability. The results of the investigation of acceptability indicate that occupational therapists and relatives of individuals with disabilities found MapIt to be useful because it provided a global view and supported joint understanding of the environment. However, some concerns have also been raised: the need for a person to scan and provide technical support at a low cost for persons with major mobility or cognitive impairments, residual usability issues (complex software installation, low intuitiveness), and scan rendering (image quality and validity of measurements).

### 3.5. Intermediary for Collaboration and Decision Making between Clients and Occupational Therapists

In five of the twelve studies, the virtual environment was used as a tool to help clients understand home modification choices and facilitate communication with practitioners in the home modification process.

In three studies, 3D interior design applications were used. They supported the process of exploring a variety of potential interior designs and home modifications and allowed users to evaluate the relative benefits and challenges of these modifications before their implementation. Atwal et al. [34] explored the perceptions of occupational therapists with regard to using a VR interior design app as an assistive tool in the pre-discharge home visit process. Occupational therapists felt that the VR app had the potential to enhance clients’ understanding of home modification and enrich communication and client involvement. Although there were several suggestions for technical fine-tuning and modifications, such as adding equipment items, occupational therapists showed a positive response overall in the ease of use and the actual use of the application across a range of clinical settings. Money et al. [32] explored the perceptions of community-dwelling older adults of a 3D interior design application. The participants believed that the application served as a useful visual tool and had a clear potential to facilitate a shared understanding and partnership in care delivery. They were able to create 3D home environments; however, many issues were identified: interface considerations, size and scale, and the need to redesign the interface/functions to require less dexterity/motor skills. They also suggested that the most valuable usage scenario involved clients and practitioners jointly using a customized 3D interior design application in a face-to-face setting. In a previous study investigating occupational therapists’ perceptions in relation to 3D interior design software, the participants saw it as a useful tool that could enhance the status of OT within the healthcare profession and improve communication [33]. Although concerns have been raised, such as the need for training for familiarity with some 3D techniques and fears that it may replace home visits, most occupational therapists found the software useful as an additional tool for home visits. Furthermore, they considered that the software had the potential to facilitate home visits rather than replace them.

Aoyama et al. [35] investigated the usefulness and usability of a tablet-based augmented reality (AR) app that supports the home modification process by superimposing 3D assistive technology (AT) items onto real home environments. The participating occupational therapists reported the usefulness of the app, which facilitates client participation and collaboration in decision making for home modifications by providing visual cues (size, function, shape, etc.) about AT to be installed in the future. However, the occupational therapists also identified technical issues, such as the need to improve the fit, look, and functions of ATs in the home environment.

Unlike studies investigating cooperation and decision-making facilitation among stakeholders in the home modification process, Chandrasekera et al. [26] used AR technology as an assistive tool of a compensatory approach in client home adaptation. They developed a hybrid space within the participant’s living environment using an AR object location and information system based on visual and spatial organization. Then, they evaluated the user perceptions of older adults who had physical impairments and mild memory loss. The technology used simultaneous localization and mapping-based AR to create a hybrid space by superimposing information on the living environment to help the elderly live independently. The participants mentioned that the technology was easy to use, useful for organizing objects within the home environment, and if available, they would certainly use it to help them with their daily activities.

### 3.6. Barriers in Practice Using VR Technology

Most studies reported technical problems, such as difficulty with the navigation controls [27,28]; a lack of household objects or equipment items [27,28]; a need to improve the fit, look, and functions of assistive devices [35]; scan rendering [22]; and a need for clearer visual prompts and an alternative keyboard interface [31]. Other reported barriers were that the technology is not suitable for use by some population groups, such as the elderly [32], a lack of low-cost software support [22], a need for sufficient training to become familiar with the 3D technology, and a fear that the technology could replace home visits [33].

## 4. Discussion

This scoping review was conducted to examine the areas of the home modification process supported by VR technology, the areas of VR technology that support the home modification process, use of VR technology, perceptions of outcomes such as usability and acceptability by users, and the manifestations of barriers in practice.

The studies included in this review were either qualitative or descriptive studies to identify the problems and explore the feasibility of VR technology in the home modification process. It can be seen that VR currently requires further preparation for widespread use and future studies. The following three primary areas of VR technology for home modification support have been identified: educational tools for clients receiving home modification services and students learning home modification, tools for home environment assessment, and intermediaries for collaboration between the client and specialist. Home modification specialists in rehabilitation include an occupational therapist, physical therapist, and rehabilitation teacher responsible for determining home modification requirements and counsel throughout the environmental modification process, and a rehabilitation engineer, an architect with sufficient knowledge of disability-related accommodation and construction to develop bid-ready specifications, a contractor, or other individuals [36].

Home modification VR for educational purposes was primarily implemented as computer simulation presenting a scenario in which the user experiences environmental barriers in a virtual home environment through an avatar and solves problems by trying potential solutions. Such computer simulation software makes it possible to transfer and apply the learned information to the physical environment and to predict dangerous situations or unintended consequences that may exist in the real environment [37]. The occupational therapists and clients that participated in the study recognized the VR software as a useful tool to confirm the safety risks of the home environment and deliver information about home modification [27,28,29]. In addition, studies examining the usability of VR software for the purpose of educating students have reported that VR software promotes learning and collaboration among experts on home modification and achievement of learning goals [19]. Although many elderly and disabled people experience barriers in the home environment, the lack of clients’ awareness of home modification, and of information or media to help decision making are also factors that complicate access to home modification and the smoothness of the process [38]. In addition, education and training for home modification are not usually part of the accreditation curriculum in construction, design, rehabilitation, or OT programs, thus, learning opportunities for home modification experts are limited [38]. Therefore, VR technology has the potential to improve client awareness of the current home environment, provide key information regarding home modification, and holds sufficient value as a tool to provide innovative solutions for learning and training for home modification experts.

VR-based home measurement software is used to measure and evaluate the environment for structural changes to improve the accessibility of the client’s home and to select suitable assistive devices. In particular, the measurement of the client’s space is one of the processes necessary for the prescription and application of assistive devices, such as mobility aids (e.g., walkers and wheelchairs) or commodes. This provides a justification for the need to purchase assistive devices to clients and related parties involved in financial supporting the health and welfare systems [39]. The participants found that VR-based home measurement technology helped home measurement and potentially reduced waiting times for home visits [22,31]. Moreover, the global view supported by VR is a useful tool for enhancing the understanding of a person’s environment [22]. The key to the home improvement process is for the client to identify the environmental barriers with a professional to implement change, add assistive technology use, and promote occupational engagement [11]. Home visits are unavoidable to ensure the smooth progress of the home remediation process and to obtain optimal results, but they are a challenge for professionals and clients. Multiple home visits can lead to an inefficient use of resources, such as funds and time [40,41], and for some clients, home visits by experts can cause stress and anxiety. In addition, clients may have a negative perception of home modification as they consider that their mobility abilities are being evaluated [34,42]. Although some technical difficulties related to 3D reconstruction remain, environment measurement software using VR presents a possibility as an alternative or complement to in-person measurement of the environment through home visits.

VR can be a useful tool to help clients understand their environment modification choices and improve communication with specialists in the home modification process. A key task in the integration of healthcare and technology is to create opportunities for healthcare providers and clients to collaborate in the decision-making processes. To realize this, it is necessary for healthcare professionals to integrate and apply new technologies and services [43,44]. The three studies included in this review suggested the possibility of using a VR interior design app, which enabled the adjustments to the space to be shared and assistive devices to be prescribed with the client, as a tool to support home modification [32,33,34]. AR is a field of VR and is a computer graphic technique that synthesizes information from the real and virtual environments to create a virtual object that seems to exist in the original environment [45]. In relation to rehabilitation and home adaptation, AR programs support assistive devices or smart homes for people with physical disabilities [46]. Moreover, a mock-up AR program for cooperation and decision making in the home modification process has been developed and used [47]. Two of the studies included in this review considered AR technology. One study used tablet-based AR software to superimpose 3D assistive device items on the real home environment [35], and the other used an AR object location and information system for home adaptation and independent living of elderly clients with memory loss [26]. The participants in those studies found that AR technology enhances shared understanding and partnership between stakeholders [35] and has value as an auxiliary tool for decision making about home adaptation and independent daily activities for clients with memory loss [26]. The participants in these studies also suggested the need for fine-tuning software operations.

Most of the studies included in this review described the development of software to support home modification and investigated the usability of the software. The studies reported the need for improvement and additional studies on the technical problems of the software. Thus, VR software requires further preparation and testing before it can be implemented in the home process in general. Although the studies included in this review reported positive results in usability evaluations, a number of technical problems and modification recommendations have been raised in studies that applied software to support home modification. Many technologies that have been developed in the healthcare field are not adopted or are abandoned for various reasons [48]. While technological innovation is widely recognized as a key contributor to health, the lack of capacity of healthcare systems to support technology programs, abandonment of technology use by individuals, and difficulties with scale-up and spread are major barriers to technology non-adoption [49]. Therefore, to achieve a widespread use of the VR software for education, home measurement, and collaboration and decision making in the home modification process, it must be ensured that it meets the needs and values of users as well as solving the current technical problems, and that the software is more convenient than the existing methods and practices. In addition, it must be ensured that it provides advantages, such as increasing the value of home modification and efficient use of resources (e.g., time and funds).

### Limitations

Only 12 studies were included in this review, and although some included quantitative data such as surveys, most studies were qualitative and focused on exploring the feasibility or usability of VR technology to support home modification. Therefore, it is not possible to demonstrate broad and general aspects of the usability or the effectiveness of VR technology for this purpose. Further studies on usability, feasibility, and the effectiveness of VR technology in the current home modification support system are also required. Furthermore, as the study sample was limited to the selection strategies described in the materials and methods section, it is possible that relevant studies were missed. Future research could extend the search to include additional databases or map the field over time.

## 5. Conclusions

This review primarily explored findings in the area of VR technology applications in the home modification process and their aspects, such as usability and acceptability. VR technology has the potential to improve client awareness of home safety, provide important information on home modification, and could be used as a useful educational tool in training home modification specialists. VR-based home measurement software can be used as an alternative or complement to in-person measurements through home visits. It also has the potential to be utilized as a tool to help clients understand their home modification choices and improve communication with specialists in the home modification process. Although most of the studies showed positive responses in the usability evaluations, perceived usefulness, and acceptability, technical problems and correction recommendations were also identified according to the VR application area. Further studies on VR technologies that effectively support the home modification process by maximizing the usability and benefits of software and addressing the identified technical problems of the software should be considered.

## Figures and Tables

**Figure 1 ijerph-18-11096-f001:**
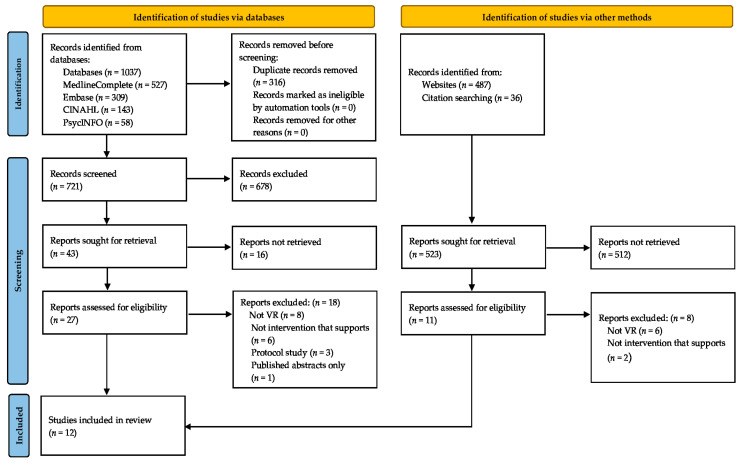
The Preferred Reporting Items for Systematic Reviews and Meta-Analyses (PRISMA) flow chart. VR: virtual reality; HM: home modification.

**Table 1 ijerph-18-11096-t001:** Summary of included studies.

Author, Year Type of Study	HM Area	Participants	Purpose	Key Findings
Sabus et al. [19] Utility study: utility questionaire and learning objective achievement analysis	Education: students	PT students (*n* = 34) OT students (*n* = 35)	To examine the utility of a web-based virtual environment for an interprofessional instructional activity	Students perceived that 3D virtual environment supported learning about home modification. 3D virtual environment facilitated interprofessional collaboration for home modifications. Learning objective achievement analysis revealed higher levels of decision making in contextual factors and appreciation of unintended consequences of recommendations.
Guay et al., 2021 [22] Usability and feasibility study: prototype development and semi-structured interview	Measurement	MapIt design: occupational therapists, engineers, clinicians, and students (total = 24) Perspective on acceptability: lay occupational therapists, older adults, and their relatives (total = 16)	To document the development and acceptability of a 3D mapping e-Health technology that allows for the exploration of home adaptations to enhance a person’s occupational engagement	Accessibility - Occupational therapists and the relatives of individuals living with disabilities felt that MapIt was useful because it provided a global view and supports joint understanding of a person’s environment. Concerns - Someone to scan and provide technical support at low cost, residual usability issues, scan rendering.
Chandrasekera et al., 2017 [26] Usability study: prototype description, usability questionnaire, and post task interview	Collaboration and decision making	Older adults with physical impairment and mild memory loss (*n* = 3)	To develop a hybrid space within the participants’ living environments using an AR object location and information system based on visual and spatial organization and to assess the users’ perceptions of using such technology	Perceived usefulness - Participants reported that a system such as this would be very helpful. Perceived ease of use - Participants mentioned that the system was easy to use. Attitude toward technology use - Participants mentioned that if this system was made available to them, they would definitely use it to help them with their daily activities. TAM questions: 7/7 (1, strongly disagree—7, strongly agree)
Threapleton et al., 2016 [27] Feasibility study: semi-structured interview	Education: stroke survivors	Occupational therapists (*n* = 13) Stroke patients (*n* = 8) Community stroke survivors (*n* = 4)	To explore perceived acceptability, potential utility, and limitations of a VR home for use in pre-discharge education and assessment	Therapists and patients perceived VR as acceptable in identifying safety risks and providing information about the need for assistive equipment and adaptations in the home. Usability issues were identified with the current technical configuration (difficulty with navigation controls, containing less general household objects, low performance of the tablet device).
Palmon et al., 2004 [28] Usability study: prototype description and observation of software trials and task completion	Education: people with physical disabilities	Occupational therapists (*n* = 8)	To evaluate usability of Habi-Test to facilitate the planning, design, and assessment of optimal home and work settings	No one completed the task on the first trial; 5/8 completed the modified task. Concerns - Frustration with the narrow field of view. - Difficulty navigating in the environment. - Difficulty recognizing the current location.
Money et al., 2019 [29] Usability study: usability questionnaire and interviews	Education: older adults	Older adults (*n* = 15)	To explore the usability of a hazard detection game and perceptions and attitudes of older adults toward using the game in practice	Good level of usability (average SUS score 77.5/100) Findings from post-task interview: - Improved awareness of home hazard detection. - Useful game as a reminder of safe practices and positive learning experience, real-world carryover effect.
Tsai et al., 2019 [30] Feasibility and usability study: questionnaires	Measurement	Lay participants (*n* = 43) Clinicians (*n* = 9)	To determine the feasibility and usability of the MPMA with an LDM that creates floor plans with detailed home measurements	77% of the lay participants spent <60 min completing the FPs. Clinician completed the virtual home evaluations within 5 min in 73% of the cases, including determining the need for DME and whether the DME is fit for the designated place. Both participant groups felt that using MPMA for home evaluations was useful and recommended it. Ease of use of the MPMA received neutral ratings from both participant groups.
Hamm et al., 2017 [31] Usability and feasibility study: usability questionnaire, analysis of think-aloud responses, and focus group	Measurement	Older adults (*n* = 33)	To explore the perceptions of older adults with regard to the barriers and opportunities of using a 3D MAP app that facilitated client-led self-assessment in the assistive equipment provision process	Good levels of usability (average SUS score 65.8/100), strong agreement with items relating to the usability (*p* = 0.004) and learnability (*p* < 0.001) Findings from post-task interview: - A useful tool to enhance visualization of measurement guidance and to promote independent living, ownership of care, and potentially reduce waiting times. - The need for clearer visual prompts and alternative keyboard interface for measurement entry is identified.
Money et al., 2015 [32] Usability and feasibility study: analysis of think-aloud responses and semi-structured interview	Collaboration and decision making	Older adults (*n* = 10)	To explore the perceptions of community-dwelling older adults with regard to adopting and using 3D interior design application as an assistive tool for the home modification process	Perceived usefulness - Participants believed the 3D interior design application served as a useful visual tool and saw clear potential to facilitate shared understanding and partnership in care delivery. Perceived ease of use - Participants were able to create 3D home environments; however, several usability issues must still be addressed. Use - Many participants did not feel confident or see sufficient value in using the application autonomously. - Most likely usage scenario would be collaborative involving the patient and practitioner.
Atwal et al., 2013 [33] Feasibility study: focus group	Collaboration and decision making	Occupational therapists (*n* = 25)	To examine occupational therapists’ perceptions of the clinical utility of 3D interior design software with the ultimate goal of exploring whether it would be a useful tool specifically for pre-discharge home visits	Occupational therapists suggested that the software could be used in discharge planning and in rehabilitation. Occupational therapists viewed it as a tool that could enhance the status of OT within the health care profession and improve communication. Occupational therapists considered the software a useful additional tool and not a replacement of expert home visits, potentially facilitating OT home visits to be more focused. Concerns - Need for training to become familiar with the 3D technology - Fear that the technology could replace home visits.
Atwal et al., 2014 [34] Feasibility study: analysis of think-aloud responses and semi-structured interview	Collaboration and decision making	Occupational therapists (*n* = 7)	To explore the perceptions of occupational therapists with regard to using VR interior design app as an assistive tool within the pre-discharge home visit process	Participants felt that the VR app had the potential to enhance clients’ understanding of home modification and enrich communication and client involvement. Perceived ease of use: all participants used the software and completed the tasks successfully. Perception of use: positive responses regarding the use of the application across a range of clinical settings Participants suggested the need for specialist equipment items to be added to the furniture library in the app and for technical fine-tuning.
Aoyama et al., 2020 [35] Usability study: Prototype description, analysis of think-aloud responses, and semi-structured interview	Collaboration and decision making	Occupational therapists (*n* = 10)	To investigate the usefulness and usability of a tablet-based AR app that supports home modification process by superimposing 3D AT items onto real home environments	Usefulness of the home modification AR: - Providing visual cues (AT fit, size, function, and appearance) in the home - Supporting collaborative home modification decision-making processes - Facilitating a holistic home modification approach - Involving stakeholders throughout the home modification processes Needs for improving the fit, look, and functions of ATs in the home environment.

HM: home modification, OT: occupational therapy, PT: physical therapy, VR: virtual reality, app: application, 3D: 3 dimensions, VR: virtual reality, SUS: systems usability scale, FP: floor plan, DEM: durable medical equipment, MPMA: MagicPlan mobile application, LDM: laser distance measurer, AR: augmented reality, AT: assistive technology, PSSUQ: Post-Study System Usability Questionnaire, MapIt: Mobile App, CIDAs: 3D Interior Design Applications, TAM: Technology Acceptance Model.

## Data Availability

The data that support the findings of this study are available from the corresponding author, upon reasonable request.

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
