# Peer review of "Use of Virtual Reality Technology to Support the Home Modification Process: A Scoping Review"

_ijerph, 2021, doi:10.3390/ijerph182111096_

Round 1
Reviewer 1 Report
This scoping review aims to examine the areas and implementation methods of home modification processes supported by VR technology. Its main contribution is that VR technology has the potential to be useful in the home modification process but technical problems of the software needs to be considered as well as usability and acceptability from the users.
This is an interesting paper and very much needed since worldwide, technology is being seen as having the potential to secure higher quality in health care and at the same time saving costs. Therefore we need these kinds of reviews that summarize the state of knowledge. A weakness of this paper is that only 12 articles were found, most of them with few participants, so consequently conclusions must be made with caution.
Keywords: home modification; virtual reality; occupational therapist
Consider revising the keywords since both home modification and virtual reality is already included in the titel
Introduction:
The introduction place the study in a broad context and highlight why it is important as well as define the purpose of the work. It seems from the introduction that it is only occupational therapist that works with home modification or is there other health professionals involved? In that case this should be clarified in the introduction. Have some minor comments:
Page 1, line 44-45. Here it mentions that home modification services involve multiple stakeholders. Could you please provide some examples of who these stakeholders might be since you further down mentions that there is a lack of communication between stakeholders involved.
Page 2, line 64. Missing a reference for this statement - VR has been gradually expanded and applied to various fields, such as defense, medical care, architecture, and education, and related content is being actively developed.
Material and methods.
The review follows the PRISMA guidelines but the materials and methods section should benefit from more detailed descriptions to allow others to replicate and build on published results. See my questions and comments below:
- What was the reason for choosing studies published between 2002 and 2021?
- How was the 1037 hits distributed between the four databases?
- What does “records identified through other sources” (Figure 1) means and how did you identify them?
- Page 3, line 142-144 is says “A qualitative synthesis of the studies selected for this review was conducted. A tabular form was organized to compare the extracted thematic information from each study, and after analyzing the themes and barriers, textual descriptions were created for each study.” Could you please expand on how identification and extraction of thematic information was done and how you created textual descriptions?
- In Figure 1 you say “studies included in quantitative synthesis”. On page 3, line 142 you say that you made a qualitative synthesis. Please clarify this.
Results:
Overall the results provides a summarized description of the scoping review. It is interesting to read although there are some parts that needs some clarifications.
Table 1. Summary of included studies needs some clarification and editing
In the column labeled Author, year and type of study it is not clear how you have categorized the studies in relation to type of studies. For some studies you use the wider term qualitative for other more specific such as feasibility study. When going through the articles included in the review I also found that for instance Atwal et al., 2013 [33] labeled qualitative is also a feasibility study. Why was it labeled only as qualitative? Money et al., 2019 [29] labeled usability study is a mix between qualitative and quantitative study so why labeled it as only usability? Please go through this and make the information on type of study accurate and congruent.
Heading 3.3. VR as an Educational Tool for Clients and Specialists (page 9, line 174). Could you clarify what you mean by clients and specialist? In the text you write for example older adults, occupational therapist and physiotherapist students, users.
Same question regarding heading 3.5. Intermediary for Collaboration and Decision-Making Between Clients and Specialists (page 9, 254). What do you mean by clients and who is the specialists? Needs clarification.
Page 9, line 258-259. “3D interior design applications support the process of exploring a variety of potential interior designs and home modifications and allow users to evaluate the relative benefits and challenges of these modifications before implementation”. Not sure if this statement needs a reference or if it is a part of your results. Please clarify.
Page 10, line 301. The heading 3.6. Barriers. Could you be more specific on what you mean? Barriers using VR technologies? Barriers in practice using VR technologies? Or?
Discussion.
The authors discuss the results and how it can be interpreted in perspective of previous studies. One thing that could be highlighted more in the discussion is the challenge of developing user-friendly Virtual Reality Technology that can be implemented in practice. Should healthcare professionals and clients be more involved in the developmental process, work together with the technology experts? What about that users does not feel confident or see sufficient value in using the application autonomously? Who will provide technical support when needed? Raises ethical issues.
Another thing is that throughout the discussion clear statement are made such as “VR technology can improve the client's awareness of the current home environment, provide information about home modification, and has sufficient value as a tool to provide a new type of learning and training for home modification experts” (line 337-339). Does this scoping review really show that VR technology can…? Shouldn’t it be might… have the potential..? This review only consist of a few studies, mainly qualitative designed and with few participants. It is still an area where more research is needed before we can be sure about its outcome. Please go through these kind of statement in the discussion and reflect on if they are in line with conclusions possible to make based on the scoping review.
Page 10, line 314. Here it says that the included studies were either qualitative or pilot studies. Categorizing included studies as pilot studies has not been done before in this study so what is the reason for describing them as pilot studies here? Further, a pilot study can have a qualitative design so it not either / or. My question here referees back to figure 1 and Type of studies.
Page 11, line335-336 it says that education and training for environmental modification are not included in the OT accreditation curriculum. This is a very general statement based on one reference and in some countries training for environmental modification is included. Please clarify this.
Page 12, line 391-392. A general statement that need some clarification. Could you provide some examples of technologies that have been developed in the healthcare field and that are not adopted or obsolete for various reasons?
A methodological discussion is missing around choice of databases, exclusion and inclusion criteria, search terms and combinations etc. What significance does the choices made have for the results?
Conclusions
Since here the “home modification specialists” are mention again it is really important to clarify what this means in this paper. Since home modification is a worldwide used method in rehabilitation home modification specialists can have different meaning in different countries.
Author Response
This scoping review aims to examine the areas and implementation methods of home modification processes supported by VR technology. Its main contribution is that VR technology has the potential to be useful in the home modification process but technical problems of the software needs to be considered as well as usability and acceptability from the users. This is an interesting paper and very much needed since worldwide, technology is being seen as having the potential to secure higher quality in health care and at the same time saving costs. Therefore we need these kinds of reviews that summarize the state of knowledge. A weakness of this paper is that only 12 articles were found, most of them with few participants, so consequently conclusions must be made with caution.
Keywords: home modification; virtual reality; occupational therapist
Consider revising the keywords since both home modification and virtual reality is already included in the title.
Response: As suggested, we revised the keywords as follows.
Keywords: collaboration; education; home environment measurement; occupational therapist
Introduction:
The introduction place the study in a broad context and highlight why it is important as well as define the purpose of the work. It seems from the introduction that it is only occupational therapist that works with home modification or is there other health professionals involved? In that case this should be clarified in the introduction. Have some minor comments:
Page 1, line 44-45. Here it mentions that home modification services involve multiple stakeholders. Could you please provide some examples of who these stakeholders might be since you further down mentions that there is a lack of communication between stakeholders involved.
Response: As suggested, we have further mentioned the stakeholder.
Page 2, line 64. Missing a reference for this statement - VR has been gradually expanded and applied to various fields, such as defense, medical care, architecture, and education, and related content is being actively developed.
Response: We inserted reference.
Material and methods.
The review follows the PRISMA guidelines but the materials and methods section should benefit from more detailed descriptions to allow others to replicate and build on published results. See my questions and comments below:
What was the reason for choosing studies published between 2002 and 2021?
Response: Since the early 2000s, practical virtual reality-based home modification related studies have started, so we set the past 20 years from this year. If you think it needs a description, it will be reflected in the 2nd revision.
How was the 1037 hits distributed between the four databases?
Response: We changed the PRISMA flow diagram in Figure 1 to the new version of PRISMA 2020, and inserted the number of literature retrieved from each database and other sources in detail.
What does “records identified through other sources” (Figure 1) means and how did you identify them?
Response: We changed Figure 1 to PRISMA 2020, and inserted the number of documents extracted through websites and citation searching.
Page 3, line 142-144 is says “A qualitative synthesis of the studies selected for this review was conducted. A tabular form was organized to compare the extracted thematic information from each study, and after analyzing the themes and barriers, textual descriptions were created for each study.” Could you please expand on how identification and extraction of thematic information was done and how you created textual descriptions?
Response: As suggested, we revised it.
In Figure 1 you say “studies included in quantitative synthesis”. On page 3, line 142 you say that you made a qualitative synthesis. Please clarify this.
Response: We changed Figure 1 to PRISMA 2020.
Results:
Overall the results provides a summarized description of the scoping review. It is interesting to read although there are some parts that needs some clarifications.
Table 1. Summary of included studies needs some clarification and editing
In the column labeled Author, year and type of study it is not clear how you have categorized the studies in relation to type of studies. For some studies you use the wider term qualitative for other more specific such as feasibility study. When going through the articles included in the review I also found that for instance Atwal et al., 2013 [33] labeled qualitative is also a feasibility study. Why was it labeled only as qualitative? Money et al., 2019 [29] labeled usability study is a mix between qualitative and quantitative study so why labeled it as only usability? Please go through this and make the information on type of study accurate and congruent.
Response: We revised what you suggested. The study type was revised in all studies with specific terms, and detailed methods were also added.
Heading 3.3. VR as an Educational Tool for Clients and Specialists (page 9, line 174). Could you clarify what you mean by clients and specialist? In the text you write for example older adults, occupational therapist and physiotherapist students, users.
Response: As suggested, we revised it. Also, the specific subject of education in the HM area of the Table 1 was mentioned.
Same question regarding heading 3.5. Intermediary for Collaboration and Decision-Making Between Clients and Specialists (page 9, 254). What do you mean by clients and who is the specialists? Needs clarification.
Response: As suggested, we revised it.
Page 9, line 258-259. “3D interior design applications support the process of exploring a variety of potential interior designs and home modifications and allow users to evaluate the relative benefits and challenges of these modifications before implementation”. Not sure if this statement needs a reference or if it is a part of your results. Please clarify.
Response: 3 of the 5 studies included in this area used 3D interior design applications, and the above description was derived as a result of the study. As suggested, we explicitly stated that the above was obtained from three studies.
Page 10, line 301. The heading 3.6. Barriers. Could you be more specific on what you mean? Barriers using VR technologies? Barriers in practice using VR technologies? Or?
Response: Reflecting your opinion, we revised it to 'Barriers in practice using VR technologies', which includes technical difficulties of VR and difficulties in practical application.
Discussion.
The authors discuss the results and how it can be interpreted in perspective of previous studies. One thing that could be highlighted more in the discussion is the challenge of developing user-friendly Virtual Reality Technology that can be implemented in practice. Should healthcare professionals and clients be more involved in the developmental process, work together with the technology experts? What about that users does not feel confident or see sufficient value in using the application autonomously? Who will provide technical support when needed? Raises ethical issues.
Response: Thanks for your comments. VR technology is one of the methods to support home modification, and it is one of the means of choice for clients and professionals involved in home modification. As you mentioned, VR technology cannot be a meaningful and satisfactory means for all clients and professionals involved in the home modification process. However, the current application of technology is an approach to solve the limitations and difficulties of existing practices, and most of the studies included in this review are studies on the potential of VR technology for home modification process. Ethical issues may arise, but we hope that you consider this review in terms of finding better alternatives compared to existing practices.
Another thing is that throughout the discussion clear statement are made such as “VR technology can improve the client's awareness of the current home environment, provide information about home modification, and has sufficient value as a tool to provide a new type of learning and training for home modification experts” (line 337-339). Does this scoping review really show that VR technology can…? Shouldn’t it be might… have the potential..? This review only consist of a few studies, mainly qualitative designed and with few participants. It is still an area where more research is needed before we can be sure about its outcome. Please go through these kind of statement in the discussion and reflect on if they are in line with conclusions possible to make based on the scoping review.
Response: As suggested, we revised the sentences (‘VR technology can…’) to the statement of the potential of VR technology in discussion and conclusions.
Page 10, line 314. Here it says that the included studies were either qualitative or pilot studies. Categorizing included studies as pilot studies has not been done before in this study so what is the reason for describing them as pilot studies here? Further, a pilot study can have a qualitative design so it not either / or. My question here referees back to figure 1 and Type of studies.
Response: We revised what you mentioned as follows.
‘The studies included in this review were either qualitative or descriptive studies to identify…’
Page 11, line335-336 it says that education and training for environmental modification are not included in the OT accreditation curriculum. This is a very general statement based on one reference and in some countries training for environmental modification is included. Please clarify this.
Response: First of all, there was a confusion in word choice, so we corrected it. ‘environmental modification’ was changed to ‘home modification’ (reconfirmed from the reference). As you mentioned, some countries include it as an OT accreditation curriculum, but it is not part of the general curriculum. In addition, in some countries, after completing the OT accreditation curriculum, it is addressed as an in-depth course such as a home improvement expert training course. Therefore, we revised this sentence to a sentence meaning that training for home modification is not being addressed as a general construction, design, or OT accreditation curriculum. We revised it as follows. Please let me know if you think further revision is needed. It will be reflected in the second revision.
In addition, education and training for home modification are not usually part of the accreditation curriculum curriculum in construction, design, rehabilitation or OT program, so learning opportunities for home modification experts are limited [37].
Page 12, line 391-392. A general statement that need some clarification. Could you provide some examples of technologies that have been developed in the healthcare field and that are not adopted or obsolete for various reasons?
Response: As you suggested, we described a few reasons for not adopting the technology. If you think additional description is necessary, please let us know, and we will reflect this in the second revision.
A methodological discussion is missing around choice of databases, exclusion and inclusion criteria, search terms and combinations etc. What significance does the choices made have for the results?
Response: Thank you for your comments to improve the quality and completeness of this review. But if methodological discussion is not necessarily required in this discussion section, could you consider it again?
If you give the same opinion in the 2nd revision, we will reflect it.
Conclusions
Since here the “home modification specialists” are mention again it is really important to clarify what this means in this paper. Since home modification is a worldwide used method in rehabilitation home modification specialists can have different meaning in different countries.
Response: As suggested, we described home modification specialists in the discussion section.
Reviewer 2 Report
The topic of the paper is of interest, and the work / the efforts the authors have invested in are well documented. The authors provide a good overview of VR technology applications in the home modification process. However, there are some weaknesses in the presentation.
1. Tables cover more than one page, but labels are not shown on all pages.
1. Check the correct reference format according to the author guidelines of this journal.
3. It is better not to use abbreviations for terms that are not mentioned again.
4. Check for format consistency in each cell of the table, and check for missing abbreviations under the table.
5. Were there any barriers to the cybersickness effect or commonly reported in VR programs and negative attitudes toward the use of technology by elderly participants? If there are any reported results for it, further describe 'barriers'.
Author Response
The topic of the paper is of interest, and the work / the efforts the authors have invested in are well documented. The authors provide a good overview of VR technology applications in the home modification process. However, there are some weaknesses in the presentation.
Tables cover more than one page, but labels are not shown on all pages.
Response: As suggested, we revised the Table 1.
- Check the correct reference format according to the author guidelines of this journal.
Response: we double-checked and revised the references.
- It is better not to use abbreviations for terms that are not mentioned again.
Response: As suggested, we checked and revised the use of abbreviation throughout the text.
- Check for format consistency in each cell of the table, and check for missing abbreviations under the table.
Response: Some of the table cells and abbreviations were revised.
- Were there any barriers to the cybersickness effect or commonly reported in VR programs and negative attitudes toward the use of technology by elderly participants? If there are any reported results for it, further describe 'barriers'.
Response: Thanks for your comments. Side effects such as cybersickness are mainly reported in fully immersive VR programs, and the VR software used in the studies included in this review were not fully immersive type, so there were no reported side effects. Decreased confidence in using technology was reported in elderly participants, and it was already mentioned in the ‘Barrier’ section.
Reviewer 3 Report
I found the study very interesting and innovating. This study tried to examine the different VR-assisted home modification services in reducing costs and increasing benefits of the interventions. The use of VR in OT has been studied many years but I thought that a scoping approach should be made long time ago to establish proper quality analysis.
The use of the scoping review method allows new ways to examine the actual perspective for its quality analysis in further studies. However, the conclusions extracted by this analysis can't be conclusive enough as there are many variables not contemplated.
I suggest the following changes in the study for a better understanding:
P.12. To include explicitly a "limitations of the study" section. It is important to include these issues due to limitations of the scoping in terms of measuring the quality of the studies and to specify the subjective point of view of the statements in the discussion.
P.4 To update Figure 1 PRISMA flow diagram, as there is a new diagram since 2020.
Author Response
P.12. To include explicitly a "limitations of the study" section. It is important to include these issues due to limitations of the scoping in terms of measuring the quality of the studies and to specify the subjective point of view of the statements in the discussion.
Response: As suggested, we added a ‘Limitations’ section.
P.4 To update Figure 1 PRISMA flow diagram, as there is a new diagram since 2020.
Response: As suggested, we revised the Figure 1.
Round 2
Reviewer 1 Report
The manuscript has improved and needed changes has been mede in an accurate way. Have just one more question, in the text you use the word physical therapist, should it be physiotherapist?
Author Response
The manuscript has improved and needed changes has been mede in an accurate way. Have just one more question, in the text you use the word physical therapist, should it be physiotherapist?
Response: Thank you for your review.
It is a word that refers to the same professional, but is used slightly differently in countries as a physical therapist or physiotherapist. In the literature referenced in this review, 'physical therapist' is used, and it is also called a 'physical therapist' rather than a 'physiotherapist' in my country. Please reconsider using the word 'physical therapist'.